# Methods to Assess Fat Mass in Infants and Young Children: A Comparative Study Using Skinfold Thickness and Air-Displacement Plethysmography

**DOI:** 10.3390/life11020075

**Published:** 2021-01-20

**Authors:** Stefanie M. P. Kouwenhoven, Nadja Antl, Jos W. R. Twisk, Berthold V. Koletzko, Martijn J. J. Finken, Johannes B. van Goudoever

**Affiliations:** 1Department of Pediatrics, Emma Children’s Hospital, Amsterdam UMC, Vrije Universiteit Amsterdam, 1081 Amsterdam, The Netherlands; h.vangoudoever@amsterdamumc.nl; 2Department of Pediatrics, Division Metabolic and Nutritional Medicine, Dr. von Hauner Children’s Hospital, LMU University Hospitals, LMU–Ludwig-Maximilians-Universität Munich, 80337 Munich, Germany; Nadja.Antl@med.uni-muenchen.de (N.A.); berthold.koletzko@med.uni-muenchen.de (B.V.K.); 3Department of Epidemiology and Biostatistics, Amsterdam UMC, Vrije Universiteit Amsterdam, 1105 Amsterdam, The Netherlands; jwr.twisk@amsterdamumc.nl; 4Department of Pediatric Endocrinology, Emma Children’s Hospital, Amsterdam UMC, Vrije Universiteit Amsterdam, 1081 Amsterdam, The Netherlands; m.finken@amsterdamumc.nl

**Keywords:** growth, body composition, body fat, fat mass percentage, adiposity, anthropometric model, body composition assessment

## Abstract

Background: Traditionally, fat mass is estimated using anthropometric models. Air-displacement plethysmography (ADP) is a relatively new technique for determining fat mass. There is limited information on the agreement between these methods in infants and young children. Therefore we aimed to longitudinally compare fat mass percentage values predicted from skinfold thicknesses (SFTs) and ADP in healthy infants and young children. Methods: Anthropometry and body composition were determined at the ages of 1, 4, and 6 months and 2 years. We quantified the agreement between the two methods using the Bland–Altman procedure, linear mixed-model analysis, and intra-class correlation coefficients (ICC). Results: During the first 6 months of life, fat mass% predicted with SFT was significantly different from that measured with ADP in healthy, term-born infants (n = 245). ICCs ranged from 0.33 (at 2 years of age) and 0.47 (at 4 months of age). Although the mean difference (bias) between the methods was low, the Bland–Altman plots showed proportional differences at all ages with wide limits of agreement. Conclusions: There is poor agreement between ADP and SFTs for estimating fat mass in infancy or early childhood. The amount of body fat was found to influence the agreement between the methods.

## 1. Introduction

Childhood overweight and obesity are major public health concerns. Childhood obesity is a risk factor for later obesity, type 2 diabetes mellitus, hypertension, dyslipidemia, and cardiovascular diseases [1,2,3,4,5,6], which form a cluster of features called metabolic syndrome. Early identification of the risk of developing metabolic syndrome is of paramount importance [7]. Rapid postnatal weight gain is highly predictive of metabolic syndrome features [8]. This may at least in part be explained by tracking of body adiposity from infancy into childhood [9] and from childhood into adulthood [10,11]. Therefore, monitoring of infant body composition might offer an opportunity to trace children at risk of future metabolic syndrome, and to target them with interventions aimed at reducing this risk.

Several methods are available for the assessment of body composition in children up to 2 years of age. Traditionally, anthropometric prediction equations using skinfold thicknesses have been used to assess fat mass, but in recent years, various methods have been developed and have largely replaced anthropometric models. Among these methods is air-displacement plethysmography (ADP), which is now considered the most accurate and validated method of determining body composition in infants and young children [12,13,14,15,16,17].

Both the PEA POD for infants and BOD POD for young children have been validated against reference values obtained from the gold-standard four-compartment model based on total body water, bone mineral content, and total body potassium [16,17]. However, no studies have longitudinally compared ADP to anthropometric prediction equations using measurements of skinfold thickness from infancy up until early childhood [18,19,20,21]. Thus, the objective of this study was to compare estimated fat mass% measured using skinfold thicknesses with ADP at different time points during the first 2 years of life in healthy, term-born infants.

## 2. Materials and Methods

### 2.1. Study Population

A nutritional intervention study was conducted as a double-blind, randomized controlled trial (ProtEUs study) at two centers: Amsterdam UMC, VU University Medical Center, Amsterdam, the Netherlands (study centre 1), and Dr. von Hauner Children’s Hospital, LMU–Ludwig-Maximilians-Universität, Munich, Germany (study centre 2). Healthy, term-born, formula-fed (n = 178) and breast-fed (n = 67) infants were enrolled between 22 October 2014, and 29 December 2016 [22]. The trial was approved by the Institutional Review Boards of VU University Medical Center Amsterdam and the Medical Faculty of LMU Munich. The study was conducted according to ICH-GCP principles and in compliance with the principles of the Declaration of Helsinki. Written informed consent was obtained from the parents or legal guardians of all participants.

The intervention period was 6 months (from an average age of 1 month until the age of 6 months). During this period, infants visited the study site three times: at 1 month (baseline), 4 months, and 6 months of age, as well as a follow-up that was scheduled at the age of 2 years. During these visits, anthropometry and body composition measurements were obtained.

### 2.2. Procedures

#### 2.2.1. Anthropometrics

Weight was measured without clothing or diaper to 0.5-g accuracy on a balance scale (MARSDEN, Rotherham, UK). Supine length was measured using a flexible measurement board (SECA, Birmingham, UK) to the nearest 0.1 cm. At the age of 2 years, standing height was measured using a digital stadiometer (SECA, Birmingham, UK) to the nearest 0.1 cm. The average of two measurements was used in the statistical analysis.

The thicknesses of the triceps, biceps, suprailiac, and subscapular skinfolds were measured in duplicate on the left side of the body to the nearest 0.2 mm using Holtain skinfold callipers (HoltainLtd, Bryberian, UK). The average of the measurements was used for the statistical analysis. The measurements were performed using a standardized protocol by well-trained observers, including research nurses and dieticians.

We used the equations of Weststate and Deurenberg [23] to estimate fat mass% from skinfold thicknesses (SFTs). The choice of this equation was based upon its applicability in the age range of 0–2 years. A validation study involving infants aged 0–4 months compared various anthropometric models to DXA for the prediction of fat mass%, and the results showed that the Weststate and Deurenberg equation had the highest R^2^ (0.77) [24].

The predicted density (D) was converted to fat mass% using the sum of bicipital, tricipital, suprailiacal, and subscapular skinfold thicknesses (⅀SFT) [23].
(1)$$\mathrm{FM}\%=\;\left(\left\{585-4.7\;\left[\mathrm{age}\;\left(\mathrm{mo}\right)\right]{\;}^{0.5}\right\}/D\right)-\;\left\{550-5.1\;\left[\mathrm{age}\;\left(\mathrm{mo}\right)\right]{\;}^{0.5}\right\}$$
(2)$$D=\left\{1.1235+\left(0.0016\;\left[\mathrm{age}\;\left(\mathrm{mo}\right)\right]{\;}^{0.5}\right)\right\}-0.0719*\log\left(⅀\mathrm{SFT}\right)$$

We used inter-rater reliability tests to estimate the reliability between the different observers. For this purpose, we organized meetings at the start of the study and throughout. During these three meetings, the observers involved performed measurements of skinfold thickness sequentially on children aged 6 months to 2 years.

#### 2.2.2. Air-Displacement Plethysmography (ADP)

Body composition was measured using a PEA POD (Infant Body Composition System; Cosmed, Concord, CA, USA) at the ages of 1, 4, and 6 months. At the age of 2 years, body composition was measured using a BOD POD (Pediatric Option Body Composition System; Cosmed, Concord, CA, USA). ADP is a densitometric technique that uses the inverse relationship between pressure and volume to calculate body density [14].

With this technique, body mass and body volume are measured. Mass is measured on an electronic scale, and volume is measured in an enclosed chamber by applying gas laws that relate pressure changes to volumes of air in the chamber. The body mass and body volume measurements are used to calculate the whole-body density using a classic densitometric approach and age- and sex-specific fat-free mass density values [25] (assuming a fixed density of fat of 0.9007 g/mL [26]). Body density is then used in a two-compartment model to calculate fat%, fat mass, and fat free mass [14,15,16].

Weight, length, and SFTs were measured first, followed by ADP. The measurements took place in a closed room with a controlled temperature and stable airway pressure according to the manufacturer’s recommendations and guidelines. The PODs were calibrated before each use by a menu-operated system according to the manufacturer’s guidelines. Body temperature and hair significantly affect the determination of fat mass% [27,28], so participants either wore a cap, or we smoothed their hair using baby oil prior the measurement. All measurements (weight, length, SFT and ADP) were performed sequentially during the same visit under identical conditions.

### 2.3. Statistical Analyses

We compared the two methods in three different ways. First of all, we used the Bland–Altman procedure to quantify the agreement between the two quantitative measurements by studying the mean difference (bias) and constructing limits of agreement [29]. The limits of agreement were calculated using the mean and standard deviation of the differences between two measurements. We set a maximum allowed difference/bias of 10% of the mean fat mass% at a particular age using reference data based on a multicomponent model of body composition of infants and young children [30]. Second, we analyzed the differences in fat mass% between the methods with linear mixed-model analysis. The alpha level was set at *p* < 0.05. All outcomes were normally distributed.

Third, the intra-class correlation coefficient (ICC) was calculated as a reliability index (in total and per centre). ICC values less than 0.5 are indicative of poor reliability, values between 0.5 and 0.75 indicate moderate reliability, values between 0.75 and 0.9 indicate good reliably, and values greater than 0.90 indicate excellent reliability. To estimate the ICC, we used a two-way mixed effects model, type single-measures, and absolute agreement. To estimate the ICC between the observers, we used a two-way mixed effects model, type average-measures, and absolute agreement. The statistical analyses were performed in SPSS (version 26).

## 3. Results

### 3.1. Subject Characteristics

We investigated 245 infants at age 1 month, which declined to 173 children at age 2 years. The majority (72–73%) were formula-fed for the first 6 months of life (Table 1). Figure 1 presents the fat mass% obtained from SFTs and measured by ADP. We set a maximum allowed difference/bias of 10% of the mean fat mass%. This comes down to 3 percentage points (2.6% at 0 months, 3.1% at 4 months, 3.1% at 6 months, and 2.6% at 2 years).

### 3.2. Comparison of the Methods

Fat mass% predicted with SFTs was significantly different from that measured by ADP except at the age of 2 years (Figure 1 and Table 2). Analyses by study site showed that the latter observation was limited to only one of the two study centers (Appendix A). Furthermore, analyses by infant sex showed that in boys at the age 4 months predicted fat mass% was no different between the methods (Appendix A). The Bland–Altman analysis revealed bias at all ages (Table 2 and Figure 2). The bias at the age of 1 month was 3.05 percentage points and was smaller at later ages. The smallest bias was found at the age of 2 years (bias 0.47). However, the limits of agreement were wider with age. The widest occurred at the age of 2 years, indicating that estimating the percentage of body fat% with SFT in children may be as much as 13 percentage points below or 14 points above the ADP measurement for an individual child.

The plots showed proportional differences in that the overestimation or underestimation of fat mass% predicted by SFTs was affected by the amount of body fat (Figure 2). All Bland–Altman plots showed an overestimation of fat mass% predicted by SFT in infants with a low body fat and the underestimation of fat mass% in infants with a high body fat. The ICCs indicated poor agreement between the two methods (range 0.33–0.47, Table 2). The highest correlations were found when comparing ADP with SFT in study center 2, for which moderate reliability was obtained (Appendix A).

### 3.3. Inter-Rater Reliability Tests–Skinfold Thickness Measurements

The inter-rater reliability tests showed high ICCs between the observers (range 0.992–0.998), indicating excellent reliability.

## 4. Discussion

This study in infants and young children showed that fat mass% predicted by SFTs is significantly different from that measured by ADP. Many anthropometric methods have been developed to estimate fat mass%. Some of them have been used in infancy, such as the non-validated but frequently used Slaughter equation. One study involving 2-week-old infants showed that fat mass% estimated from the Slaughter equation [31] was strongly correlated (ICC range 0.69–0.74) with fat mass% predicted from ADP [32]. However, we and others could not confirm these observations [33]. In our sample, ICCs ranged from 0.25 to 0.57 (data not shown).

Another method for estimating fat% in the very early stages of life is the equation presented by Catalano et al. This method uses the flank and thigh skinfolds in addition to the triceps and subscapular skinfolds [34]. The equation was shown to be reasonably accurate as compared to ADP in the estimation of fat mass in neonatal life, but not at later age [18,19].

Despite the limitations, the development of a suitable and validated anthropometric model to predict body fat in infants and young children is of great interest because it would have low cost and widespread availability. Recently, the adjustment of traditional anthropometric equations by using ADP as a reference has led to new equations to predict fat mass in infants and young children. However, a validation study showed that the outcomes were still significantly different from those of ADP [18].

Another study involving neonates and young infants compared skinfold thicknesses to DXA-derived fat mass. The results showed that an exponential equation including the sum of four skinfold thicknesses and length best predicted fat mass [24]. However, this equation was recently used in a cohort of preterm infants and showed poor agreement with DXA-derived fat mass at term-equivalent age [35].

It is frequently reported that the measurement of skinfold thicknesses in young children suffers from limited reproducibility and precision. Notably, the interobserver error is high [36]. This aspect of measurement error is one of the major criticisms of skinfold thickness measurements for estimating body fat in infancy. Even though we found high ICCs between the observers in our study, there were differences of up to 2 mm in the obtained skinfold thicknesses between observers. A difference of 2 mm for every skinfold thickness measurement adds up to a difference of 8 mm in ∑SFT. Consequently, body density will differ by 0.009 kg/L, yielding a difference of 5 percentage points in predicted body fat.

The strength of our study is the inclusion of high numbers of paired observations in a healthy population from infancy to early childhood. Furthermore, this study was performed in two experienced clinical paediatric research centres in Europe, which adds external validity. Nevertheless, an important limitation of our study is the low number of objective methods used for estimating body composition.

## 5. Conclusions

Among healthy infants and young children born at term, there is poor agreement between estimates of fat mass by SFTs and ADP. This hampers the comparability of studies using different methods for the assessment of body composition. Further research on the comparability of different body composition methods for longitudinal assessment of adiposity is warranted.

## Figures and Tables

**Figure 1 life-11-00075-f001:**
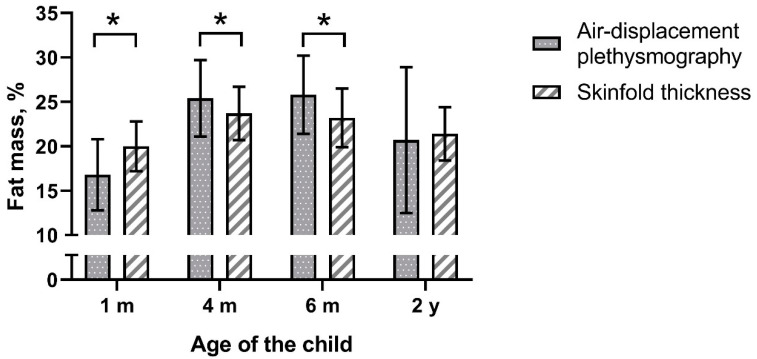
Fat mass% measured by air-displacement plethysmography and predicted with skinfold thickness, mean ± SD, * *p* < 0.001.

**Figure 2 life-11-00075-f002:**
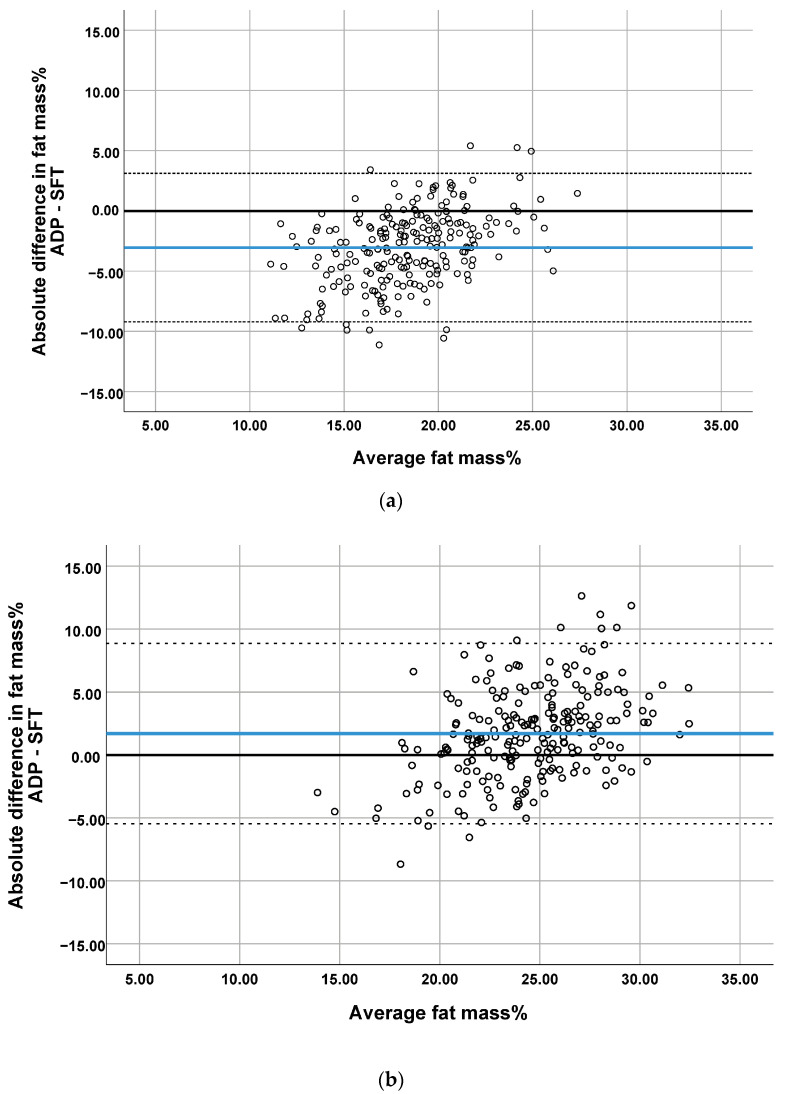
Bland–Altman plots comparing the prediction of percentage of body fat by SFT with ADP at (**a**) 1 month, (**b**) 4 months, (**c**) 6 months and (**d**) 2 years. Blue line represents the mean difference (bias). The dotted lines represent 95% Limit of Agreement.

**Table 1 life-11-00075-t001:** Infant characteristics.

	1 Month	4 Months	6 Months	2 Years
	n = 245	n = 238	n = 235	n = 173
Boys [n (%)]	113 (46)	112 (47)	111 (47)	82 (47)
Caucasian [n (%)]	213 (87)	208 (87)	205 (87)	151 (87)
Age (mo)	1.0 ± 0.3	3.9 ± 0.1	6.0 ± 0.1	24.1 ± 0.5
Body weight (gram)	4207 ± 536	6598 ± 685	7692 ± 834	12415 ± 1315
Length/height (cm)	54.5 ± 2.2	63.7 ± 2.0	67.9 ± 2.2	86.8 ± 2.8
Formula-fed ^a^ [n (%)]	178 (73)	173 (73)	170 (72)	NA
Skinfold thickness measurements				
	n = 242−235	n = 237−236	n = 235	n = 169−168
∑SFT (mm) ^b^	23.2 ± 3.8	29.4 ± 5.2	29.0 ± 5.6	27.3 ± 5.2
	n = 235	n = 235	n = 235	n = 167
Body density (kg/L)	1.0274 ± 0.0050	1.0215 ± 0.0055	1.0228 ± 0.0060	1.0286 ± 0.0057
Fat mass%	20.0 ± 2.8	23.7 ± 3.0	23.2 ± 3.3	21.4 ± 3.0
Air-displacement plethysmography				
	n = 231	n = 230	n = 219	n = 103
Body density	1.0330 ± 0.0073	1.0185 ± 0.0077	1.0184 ± 0.0079	1.0327 ± 0.0154
Fat mass%	16.8± 4.0	25.4 ± 4.3	25.8 ± 4.4	20.7 ± 8.2

Mean ± SD, ^a^ randomized into intervention or control formula, ^b^ Sum of bicipital, tricipital, subscapular, and suprailiacal skinfold.

**Table 2 life-11-00075-t002:** Agreement between the different methods for the prediction of body composition (air-displacement plethysmography (ADP) minus skinfold thickness (SFT)).

	1 Month	4 Months	6 Months	2 Years
n	225	229	219	102
Mean difference (bias)	−3.05	1.70	2.65	−0.47
Limits of agreement (CI95%)	3.12 −9.21	8.86 −546	10.25 −4.96	13.29 −14.23
Difference ^1^	−3.15	1.70	2.63	−0.67
*P* ^1^	<0.001	<0.001	<0.001	0.112
(CI95%)	(−3.75, −2.54)	(1.10, 2.31)	(2.02, 3.24)	(−1.49, 0.16)
ICC ^2^	0.42	0.47	0.42	0.33
	1 month	4 months	6 months	2 years

^1^ Differences between the methods by linear mixed-models analysis, ^2^ ICC; Intra-Class Correlation Coefficient.

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
