# Peer review of "Methods to Assess Fat Mass in Infants and Young Children: A Comparative Study Using Skinfold Thickness and Air-Displacement Plethysmography"

_life, 2021, doi:10.3390/life11020075_

Round 1

Reviewer 1 Report

Tale 1 - 72% were formula fed at 2 years. This is unusual - that is the public health guidance for milk feeding at this age, and why was the % of children on formula milk so high?

Line 216: It would be helpful to include a few lines on directions for future research in light of your findings/observations.

Author Response

Table 1 - 72% were formula fed at 2 years. This is unusual - that is the public health guidance for milk feeding at this age, and why was the % of children on formula milk so high?

We agree with the reviewer this is not clearly written in the manuscript.

This study was part of a nutritional intervention study in healthy, term-born infants. Within this nutritional intervention study we assessed the safety of a novel infant formula with a modified amino acid profile. We enrolled 245 infants. Of these infants, 178 were formula-fed and 67 were breast-fed infants at study entry. This comes down to 73% of the included infants.

The intervention started at baseline/age 1 month until the age of 6 months.

We edited our manuscript to make this more clear to the reader:
Table 1: We deleted the formula-fed numbers at the age of 2 years.

Line 216: It would be helpful to include a few lines on directions for future research in light of your findings/observations.

We agree with the reviewer and added line 225 to the manuscript.

Reviewer 2 Report

The paper is very interesting and written well with the high substantial quality. I am pleased to recommend it for publication in the present form.

Author Response

Thank you for your compliments.

Reviewer 3 Report

Please find attached my comments.

Author Response

Methods to assess fat mass in infants and young children: A comparative study using skinfold thickness and air-displacement plethysmography This study aims to assess agreement between anthropometry and air-displacement plethysmography methods for quantifying fat mass in infants and young children, with the measurements made longitudinally at 1, 4, 6, and 24 months. Paired t-test, intra-class correlation coefficients, and Bland-Altman analysis were applied for the assessment, and the authors concluded that the two methods had poor agreement. Overall, it’s a well written manuscript with clearly articulated study objective. I suggest the authors considering the following comments, which I hope help improve the manuscript.

Comments:

  • Interpretability of results in Table 2 relies on several (statistical) assumptions, including normality and homogeneity of variance. Please comment if the data was tested against these assumptions, and the sample size requirement for agreement limit assessment was achieved.

We agree with the reviewer that this is not described in the methods. We tested the normality and the data is normally distributed.
We visualized the distribution of the variables by creating a histogram for every variable. Please find the histograms of the variables in the attachment.

We added line 129 to the section Statistical Analyses.

  • For air-displacement plethysmography, body composition was measured using PEA POD at the age of 1, 4, and 6 months, and using BOD POD at the age of 24 months. Mean differences between the two methods were significant at the age of 1, 4, and 6 months, but not at the age of 24 months. It’s however not clear if that has something to do with fat mass change with age or method for measuring body composition (PEA POD versus BOD POD).

Overall we found a significant mean difference between the methods at all ages, except at the age of 2 years. However, additional analyses showed that this non-significant difference is seen in one of the two study centers only.
Even the mean difference between the methods is not significant at the age of 2 years, the low ICCs indicated poor agreement between the two methods.

The Bland Altman plot clearly shows that there is a proportional difference between the methods at the age of 2 years. With the widest limits of agreement compared to other ages. This can be caused by the methods (ADP vs. skinfold thickness), the equipment (Bod Pod) and the age category.

  • A cross-sectional Bland-Altman based analysis was applied at 1, 4, 6, and 24 months, independently, but results were interpreted as a longitudinal finding. Preforming analytic method that account for repeated measurements (and show trends with the time factor) may be needed for the comparison across time points.

We agree with the reviewer and we analyzed the differences in fat mass% between the methods with linear mixed-model analysis. We added interactions to obtain the differences between the methods at different time points.

The results of the analyses are added to table 2 and table S1.
Furthermore, we edited our manuscript by changing the type of statistical analysis, line 120-121, 128-129. We added line 152.

Next to the analyzed differences, the observed differences (bias) are presented in table 2 and table S1. The figures represents the bias between the methods.

  • In Table 1, please provide information about number of male and Caucasian infants for measurement at 4, 6, and 24 months, as for the measurement at 1 month. Additionally, I suggest presenting analyses of agreement between the two methods per infant sex. This may be added as a supplementary material.

Thank you for this valuable suggestion. The number of males and Caucasian infants per age category are added to table 1. The percentage of boys and the percentage of Caucasian infants does not differ between the ages.

The results of analyses of agreement between two methods per infant sex are presented in table S2.
We found that there is a significant difference between the methods in girls but not in boys at the age of 4 months.

We added lines 152-155 to the manuscript.
